# Surgical Management of Periprosthetic Joint Infections in Hip and Knee Megaprostheses

**DOI:** 10.3390/medicina60040583

**Published:** 2024-03-31

**Authors:** Christoph Theil, Sebastian Bockholt, Georg Gosheger, Ralf Dieckmann, Jan Schwarze, Martin Schulze, Jan Puetzler, Burkhard Moellenbeck

**Affiliations:** 1Department of Orthopedics and Tumor Orthopedics, Muenster University Hospital, Albert-Schweitzer-Campus 1, 48149 Muenster, Germany; 2Department of Orthopedics, Brüderkrankenhaus Trier, Medical Campus Trier, Nordallee 1, 54292 Trier, Germany

**Keywords:** megaprosthesis, megaprostheses, revision arthroplasty, sarcoma, metastases, hip arthroplasty, knee arthroplasty

## Abstract

Periprosthetic joint infection is a feared complication after the megaprosthetic reconstruction of oncologic and non-oncologic bone defects of including the knee or hip joint. Due to the relative rarity of these procedures, however, optimal management is debatable. Considering the expanding use of megaprostheses in revision arthroplasty and the high revision burden in orthopedic oncology, the risk of PJI is likely to increase over the coming years. In this non-systematic review article, we present and discuss current management options and the associated results focusing on studies from the last 15 years and studies from dedicated centers or study groups. The indication, surgical details and results in controlling infection are presented for debridement, antibiotics, irrigation and retention (DAIR) procedure with an exchange of the modular components, single-stage implant exchange, two-stage exchanges and ablative procedures.

## 1. Introduction

Megaprosthetic reconstruction is a commonly chosen approach following the resection of bone tumors or in revision arthroplasty with severe bone loss [1,2]. In particular, this is considered for juxta-articular bone tumors affecting the proximal and distal femur as well as the proximal tibia, given the main manifestations of malignant bone tumors [3]. Furthermore, in patients with complicated revision hip or knee arthroplasty with severe bone loss and segmental defects, modular megaprostheses have become a valuable option in addressing these complex situations. The main advantage includes implant modularity, enabling the surgeon to intraoperatively adapt the reconstruction to the individual defect as opposed to custom-made implants [3,4]. Furthermore, modern implant systems are widely available and allow for early functional rehabilitation compared to biological reconstructions [5].

With long-term experiences on the use of hip and knee mega-implants from dedicated tumor centers showing excellent results regarding long-term limb salvage and good results regarding implant survivorship [6], possible indications were expanded to non-oncologic reconstructions [7,8,9].

With the overall number of hip and knee arthroplasty procedures rising and continuously high numbers in many western societies [10], the associated number of revision procedures is expected to rise drastically over the coming years [10]. Major bone loss extending beyond the metaphysis may occur, in particular, with repeat revision surgeries, especially those facing recurrent infections or fractures, which sometimes necessitate the removal of long-stemmed implants [7,8,9]. In these situations, stable implant fixation can be compromised and segmental bone replacement becomes an option [11,12,13]. Generally speaking, the risk of re-revision in the setting of megaprosthetic reconstruction as part of a revision arthroplasty procedure is increased, but good long-term results are possible even in challenging situations [7].

The current survival for oncologic hip and knee megaprostheses has been reported to be fairly good [14,15]. One study on modern design distal femoral replacements found a revision-free survival of 58% after 8 years [16]. Likewise, another study reported an excellent survival of 75% for proximal femoral replacement in oncology patients after 10 years [17]. For revision arthroplasty procedures, long-term studies are scarce, however, one study found that, in general, the revision-free survival of a distal femoral replacement can be 73% after five years [18]. Likewise, revision total hip arthroplasty (THA) with a proximal femoral replacement can have good mid-term survival [11] of around 73% after five years.

A major unresolved issue of oncologic and non-oncologic megaprosthetic reconstructions is the risk of periprosthetic joint infection (PJI) [9,15,19,20,21,22,23]. For primary oncologic implants the general assumption is that the risk of PJI is about 1% per year, or 5–10% for studies with mid-term follow-up [6,24,25]. However, once these implants undergo revision surgery, the re-revision rate for infection can be around 20%. Similarly, megaprosthetic reconstruction in non-oncologic patients can have a revision risk for PJI of up to 50%, depending on the initial indication [9,13,26]. In these situations, management options range from non-curative, minimally invasive procedures, such as persistent fistula or antibiotic suppression, to invasive multi-stage revisions [27]. Depending on the limb’s and patient’s conditions as well as patient preference, amputation must be discussed and seriously considered as a primary or rescue treatment, particularly in uncontrolled infections [19,25,28]. The risk of amputation due to a megaprosthetic PJI has been reported to be up to 36% compared to those without PJI [25].

The majority of studies on the management of infected megaprosthesis is limited by small patient numbers and heterogeneous indications and implants [19,29]. The optimal systemic and local antibiotic treatment as well the optimal surgical management are unclear.

The aim of this review is to provide an overview on antibiotic and surgical treatment options based on the available literature as well as our own experiences.

## 2. Materials and Methods

For this review, a selective literature search was performed. The focus was to include larger studies from dedicated centers in order to include a maximum number of patients to base recommendations on. Furthermore, studies were given preference for review if they used uniform definitions based on consensus criteria, such as the criteria proposed by the MSIS (musculoskeletal infection society) [30], ICM (international consensus meeting) [31] or EBJIS (European Bone and Joint Infection Society) [32] or institutional criteria that echo the general idea of these criteria. For the assessment of megaprosthetic failures, studies that used the ISOLS (International Society on Limb Salvage) consensus classification by Henderson et al. of 2011 or the revised version of 2014 were preferred [33,34].

Management strategies were grouped by general consideration regarding local and systemic antimicrobial treatment, non-curative approaches, implant retention approaches with single-stage exchange of modular components, complete single stage implant exchange, partial two-stage exchange with retention of a well-fixed stem, complete two-stage exchange and ablative procedures.

Considering the small volume of megaprostheses implanted, particularly for oncological indications, and the resulting heterogeneity in the literature in terms of definitions, treatment protocols and time periods included, we chose not to perform a systematic review or meta-analysis as any results will be severely impaired by the aforementioned factors [35].

## 3. Results

### 3.1. Systemic and Local Antimicrobial Treatment

Systemic antibiotics are an important aspect in the management of PJI. While shorter durations of antibiotic therapy are possible, it is generally accepted that a minimum of six weeks of systemic therapy should be performed [36]. However, in recent years there is increasing high-quality evidence that twelve weeks of systemic treatment yields better infection control compared to six weeks [37]. Furthermore, prolonged oral antibiotic therapy for three months after a two-stage exchange appears to lead to a reduced risk of reinfection [38]. Considering that megaprosthetic reconstructions are at an increased risk of PJI and recurrent PJI after revision, prolonged antibiotic therapy appears warranted. Furthermore, for individual high-risk cases or situations in which revision surgery is associated with inadequate morbidity, suppressive antibiotics can be considered, although the available evidence is relatively poor [27].

On the other hand, there is increasing interest in local antibiotic therapy in order to potentially achieve better results, or reduce the duration of systemic therapy and thus, the potential side effects of some systemic antibiotics [39,40]. There are ongoing high-quality trials to address this question, although these are not focused on megaprostheses. While local antibiotic carriers are available in many different forms and from a number of manufacturers or as a custom-made product, individual application depends on various factors, such as the desired elution, planned secondary surgeries, antibiotic susceptibility and availability.

In our own practice, antibiotic-loaded polymethylmethacrylate (PMMA) spacers are inevitable in staged revisions with large bone defects, but obviously require secondary removal, and antibiotic elution from the PMMA is limited [41,42]. In the past, antibiotic collagen sponges with the addition of gentamicin were used, but there is little evidence to support their use in megaprosthetic reconstructions and relevant amounts of antibiotics might only be present for a few days [43].

More recently, calcium-based carriers have been introduced [43]. While the addition of hydroxylapatite or tricalciumphosphate allows for bone regeneration and can be used for bone defects or intramedullary, calcium sulfate-based carriers can be used in the soft tissues or even around implants when implant retention is desired [44]. We are currently investigating the initial results of salvaging megaprosthetic reconstructions with modular component exchange as part of DAIR (debridement, antibiotics, irrigation and implant retention) with the addition of calcium sulfate beads as a local antibiotic carrier. While the short-term results are quite promising, long-term comparative studies are not available [40].

### 3.2. DAIR

A DAIR procedure with a retention of the implant has the obvious advantage that the relatively high morbidity of (staged) complete implant revision can be avoided with decent infection control in around 50–70% of patients [45]. It has been investigated for early or acute infections with a follow-up timeframe of only 3–6 weeks of symptoms since the last surgery. The rationale for the chosen interval was that the formation and maturation of biofilm on the implant prohibits the successful eradication of the infection with (partial) implant retention, although there is no hard threshold. However, the optimal indication, including surgical details such as the irrigation solution, use of local antibiotics and susceptibility of the infectious organism are debated, and may greatly impact success rates [19,26,45]. It is, however, agreed that thorough debridement is one key factor [46].

For megaprosthetic PJI, the use of DAIR is even less well investigated and we have only moderate evidence to support or reject its use [19]. The reported success rates vary greatly and the reported cohorts are usually limited to up to fifteen patients [47]. Nonetheless, the same limitations with regard to diagnosis, detailed surgical management and the potential impact of microbiological factors can be applied. Still, there are also reports that show successful long-term infection control in non-oncological femoral megaprostheses in almost two-thirds of all patients [48].

Considering that, with modern modular implants, a large proportion of the implant surface can be exchanged as part of a DAIR (known as DAIR plus) and the established high morbidity of full implant exchange [9], a DAIR procedure can be a viable option with well-fixed implants. Factors that might contribute to a successful megaprosthetic DAIR are known and include susceptible organisms, good soft tissue coverage, a relatively healthy host and short duration of symptoms.

In our own experience, DAIR procedures should be individually discussed with the patients, and should reflect potential beneficial or risk factors to estimate the expected likelihood of success. As there is no good evidence [49] that a prior DAIR procedure negatively impacts the success of a staged revision, we feel that offering a DAIR to patients with megaprosthetic PJI is a viable option if the implant is stable. Furthermore, from our own experience in individual cases, DAIR plus antibiotic suppression or watchful waiting may allow limb preservation and the avoidance of further highly invasive surgeries in patients with good function, high surgical risk or short life expectancy. This might be desirable in some patients, even if the infection is not eradicated, but transferred into a controlled state.

### 3.3. Single-Stage Exchange

A single-stage complete implant exchange is an established option for the management of chronic PJI and is performed with good success, particularly in specialized centers [50,51,52]. Citak et al. demonstrated that in hip PJI even extensive proximal femur defects can be treated with a single-stage septic exchange with a low reinfection rate of 16% [50]. However, for megaprosthetic reconstruction the terminology can be confusing and some authors refer to a DAIR with an exchange of the modular components (DAIR plus) as a single-stage exchange which would actually entail exchanging the stems, even if well-fixed as well [25]. Jeys et al. showed an infection control rate of around 50% for one-stage revisions; however, in their study two-stage exchange was much more successful and resulted in an infection control rate of 72% in a similar group of patients [25].

The advantages of a single stage revision for chronic PJI are the obvious lack of a planned second surgery and the period of relative immobilization with a megaprosthetic spacer. However, available data on this approach in the presence of a megaprosthesis is scarce. Furthermore, terminology in past studies might have classified DAIR plus procedures with an exchange of modular components as a single-stage revision, complicating assessment even more. In the authors’ view, a complete single-stage revision is rarely performed and should be discussed on an individual basis if patients are not medically fit enough to undergo a two-stage approach. In our own practice, a DAIR plus procedure would be favored in such cases. If a one-stage revision is chosen, the traditional precautions with an identified organism, fair soft tissues and antibiotic-loaded cemented implant fixations must be considered as potential success factors.

### 3.4. Two-Stage Exchange

In cases with chronic infections of hip and knee megaprostheses with a duration of symptoms of more than 4–6 weeks, the removal of the implant becomes necessary to cure the PJI. While one-stage exchanges are well described for a hip and knee revision of primary arthroplasty with similar success rates, two-stage revision arthroplasty remains the gold-standard for chronic infections, especially in revision arthroplasty [53,54,55,56,57]. A two-stage approach allows for an aggressive debridement even in the presence of poor soft tissues and festering infections with the necessity to remove infected bone, as there is no need to perform a reimplantation of a new prosthesis during this surgery; however, there are several downsides to two-stage revision.

Usually, particularly in the setting of large femoral or tibial bone defects, a static spacer leading to the temporary arthrodesis of infected knee joints [9], or an articulating spacer for proximal femoral and hip infections, is used [23,58]. The spacer is usually made from rods or intramedullary nails coated with antibiotic-loaded polymethylmethacrylate bone cement and allows for stability even in the presence of the large bone defects that must be expected in patients with megaprosthetic infections. Furthermore, high dose local antibiotic delivery is possible within the first weeks. Nonetheless, mechanical complications, such as the dislocation of hip spacers, spacer breakage or fracture of the adjacent residual bone, are possible; therefore, patients must be instructed to limit their activity during the interim period and a knee brace must be worn. In order to facilitate prosthesis reimplantation, the spacer body should be fabricated to be sufficiently large and long enough, as long as the closure of the fascia and muscle coverage are possible, to allow for a later accommodation of a revision megaprosthesis at the time of reimplantation.

While long-term spacer retention can be an option in selected patients who have decent function [59], this is rarely the case in megaprosthetic infections due to the large defects usually present. Therefore, second-stage reimplantation should be performed after a course of systemic antibiotics, usually after six weeks although longer intervals are possible in everyday practice due to various patient and organizational factors. Recent data suggest that an interval of up to 90 days does not lead to worse outcomes in two-stage revisions [60]. However, in oncological patients who experience megaprosthetic PJI during chemotherapy or radiation therapy, a spacer might be retained for longer periods of time as long as there is no sign of an ongoing, persisting infection, because the completion of the adjuvant oncological treatment precedes reimplantation surgery. However, in some oncological patients with a poor prognosis, individualized treatment regimes might become necessary, e.g., in patients with progressive metastases and a retained spacer who wish to resume weight-bearing during a potentially limited residual life expectancy.

At the time of reimplantation, surgeons must be aware that a certain percentage of patients might have positive cultures at this point, although this has mainly been investigated for non-megaprosthetic two-stage exchanges. In our practice, antibiotic therapy is continued on a calculated basis and adapted if cultures are positive [61,62]. The value of serum parameters or spacer aspiration with microbiology culture and cell count is debated [63,64]. While spacer cultures show poor diagnostic performance, as do a variety of serum parameters [65], there appeared to be some usefulness for synovial fluid analysis from spacer fluid in one study on 82 two-stage exchanges for knee PJI, demonstrating a sensitivity and specificity of around 80% to diagnose persisting infection [64]. In our own practice, we do not perform spacer aspiration routinely, but it can be an option in patients who have elevated serum inflammatory markers with no other explanation.

As for implant fixation, there are, likewise, successful reports using cemented and uncemented implants, depending on multiple factors such as residual bone length, previous implant fixation, potential fractures or fissures that need to be bridged or the presence of ipsilateral hardware [29,47,66,67]. Most modern modular implant systems allow for the cemented or uncemented fixation of different length stems as well as for combination with the modular fluted stems that have proven useful in revision hip arthroplasty and can be coupled with preexisting modular components [68,69]. While we generally prefer a cementless stem fixation, particularly on the femoral side, modular coupling devices that allow for such stems are associated with a risk of implant breakage that has otherwise become rare in modern implants [33].

One large study on 83 patients with a two-stage exchange of an oncological lower extremity megaprosthesis reported a long-term infection control of 75% [70]. However, even in more recent studies, long-term infection control depending on microbiology factors, soft tissue condition, adjuvant treatments, particularly radiation, and types of reconstruction might only be around 50% [24,25,29,71]. For non-oncological megaprosthetic reconstruction, e.g., in patients with multiple previously failed arthroplasties, the results vary. While one study on 49 proximal femoral replacements performed for PJI [7] reported a 92% infection eradication with the use of a PFR, the risk of reinfection increased to 18% after revision for mechanical complications, mainly dislocations, in this cohort with mid-term follow-up. The authors cite severe comorbidity in those patients who had reinfections and multiple previously failed interventions.

For non-oncological reconstructions of the distal femur using a megaprosthesis, the reported results are fairly dismal. One study on 30 DFR performed for various non-oncological indications reported an infection risk of 27%. However, when only considering patients who had the DFR as part of PJI management, the reinfection risk was 56% in this study. Mid-term infection control with the preservation of a functioning DFR arthroplasty was only successful in 38%, with 50% of the patients who suffered from infection undergoing transfemoral amputation to cure the infection. Likewise, another study on 29 DFR reconstructions for PJI found a reinfection risk of 41% at mid-term follow-up [1]. However, management was quite different, with the majority of patients with reinfection receiving suppressive antibiotics or undergoing DAIR procedures to retain the implant.

### 3.5. Partial Two-Stage Exchange

Considering the morbidity that is potentially associated with removing a well-fixed uncemented stem [72], which might lead to relevant bone loss and a potential need to replace an additional joint (total bone replacements), it has been proposed to retain such stems in oncological patients under certain conditions [21,73,74]. Generally speaking, a complete implant removal is desirable in order to eradicate all biofilm-coated foreign surfaces; the retention of uncemented stems can yield good infection control as has been shown for infected total hip arthroplasties as well [75]. Considering the poor results of total bone replacements, particularly in cases of PJI [58], we recommend to discuss and consider a partial two-stage exchange in situations where complete implant removal will lead to excessive bone loss necessitating total bone replacements. However, in cases of infection persistence/persisting wound healing issues, secondary implant removal and spacer exchange may become necessary. This additional surgery would obviously lead to further surgery-related morbidity that should have been avoided by retaining a well-fixed stem in the first place. Considering the rarity of such procedures and the lack of high-quality comparative studies, optimal patient selection is unclear and the standard approach should remain to remove all foreign material. In the setting of megaprosthetic PJI, Sigmund et al. [29] compared different surgical procedures to treat the PJI of a megaprosthesis at a single center. They found that 64% of patients (*n* = 14) with a retained stem experienced reinfection compared to only 22% (*n* = 18) of patients with complete implant removal. They concluded that if one stem is retained, the results are comparable to single-stage component revision (DAIR plus), which should then be preferred, particularly in patients who are in poor health and unfit to undergo repeat surgeries. Nonetheless, their findings are limited by small numbers and, from our own experience, a partial two-stage exchange can lead to adequate infection control, although published results are scarce and future studies are ongoing that will potentially further elucidate that topic.

### 3.6. Ablative/Semi-Ablative Procedures

While limb salvage remains the ultimate goal in treating extremity sarcomas and arthroplasty failure, PJI can lead to the need to discuss amputation and is the second most common reason for amputation after megaprosthetic reconstruction in sarcoma surgery following local recurrence [24]. Unfortunately, recurrent PJI in non-oncologic megaprosthetic revision arthroplasty is not uncommon [9,27]. The risk of amputation varies and is highest for patients with total bone replacements and tibial megaprostheses [24,58]. In some cases with severely compromised soft tissues, such as after extra-articular resections [22], amputation might be discussed as a primary surgical approach in the treatment of PJI. However, in most patients, amputation is preceded by several failed attempts to salvage the implant and limb, particularly in those who have undergone megaprosthetic replacement following complicated revision arthroplasty [9].

In this complex patient population, particularly in those patients who still undergo systemic chemotherapy or are required to undergo further treatment in managing metastatic disease, amputation can be offered and discussed as a radical, but effective and fast approach to cure the infection in order to continue oncological treatment.

## 4. Conclusions

The optimal management for megaprosthetic PJI of lower extremity reconstructions is still debated. For early and acute infections, a DAIR procedure with the exchange of all modular components (DAIR plus) results in satisfactory infection control while for chronic infections, a two-stage exchange with removal of all implant components yields good long-term infection control. Nonetheless, the results are inferior compared to surgical management of non-megaprosthetic hip and knee PJI and limb salvage is not always possible. While there are some promising results, the role of local antibiotic carriers still needs to be determined. Future research efforts should address the lack of comparative studies of relevant size and potential basic science advancements in implant coatings.

## Data Availability

No new data were created or analyzed in this study. Data sharing is not applicable to this article.

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
