# Peer review of "Surgical Management of Periprosthetic Joint Infections in Hip and Knee Megaprostheses"

_medicina, 2024, doi:10.3390/medicina60040583_

Round 1

Reviewer 1 Report

Comments and Suggestions for Authors

The manuscript titled "Surgical management of periprosthetic joint infections in hip and knee megaprostheses" focuses on interesting topic. I commend the authors for their dedication and express my appreciation for the chance to evaluate their manuscript. The manuscript is crafted in a good manner, that can in terms on English be also improved. Congratulations to the authors on their findings. A non-systematic review that in my opining should be published. The main question addressed by the research is clear and relevant (surgical management of periprosthetic joint infections in hip and knee megaprostheses in oncological and non-oncological settings), and the topic is original and adds valuable insights to the subject area (it is a review). The conclusions are consistent with the evidence and arguments presented and effectively address the main question posed.

After assessing the manuscript, the following issues raised my concerns or represent suggestions that from my point of view can increases the overall quality of the manuscript:

-       3.3. single-stage exchange - Single-stage exchange

-       3.4. two-stage exchange - Two-stage exchange

-       3.5. partial two-stage exchange - Partial two-stage exchange

-      3.6. ablative/semi-ablative procedures - Ablative/semi-ablative procedures

Author Response

Thank you very much for your appreciative review. We have changed the spelling mistakes.

Thank you and kind regards

Reviewer 2 Report

Comments and Suggestions for Authors

Thank you for submitting this manuscript 

the main question  addressed by the review was the best valid options to manage infected megaprosthesis joint implants

the topic addresses a gap in the field with relatively sparse data about megaprosthesis failure due to infection

the data was available in literature but the contribution of the authors was to collect these pertinent data and add their experiences 

the article was better performed as systematic review with inclusion and exclusion criteria to reach an evidence based conclusion.

why did not you add discussion section in which you can add your opinion rather than putting them in the results section

there is some spelling mistakes

the conclusion is sound based on the data provided

the references are appropriate 

Comments on the Quality of English Language

Minor English editing is required

Author Response

Dear reviewer,

thank you very much for your comprehensive review. 

We agree with you that reviews are best conducted in a systematic fashion with a meta analysis, however in agreement with a recent review article by Lozano-Calderon et al. on a similar topic, we chose to to do a non-systematic review, considering the great heterogeneity and general scarcitiy of studies on that topic.

All available literature includes extremely heterogeneous patient cohorts, implant systems and definitions that might have been appropriate 10-20 years ago, but have been replaced by moden definitions as cited in our article. 

We therefore chose to present selected studies (that represent the largest and best ones available on that topic) and our own experience.

We have elaborated on that in lines 95 following.

Furthermore, the manuscript has been repeatedly spell checked and improved also including the comments of the other reviewer.

Thank you and kind regards